# Electrophysiological and Imaging Biomarkers to Evaluate Exercise Training in Patients with Neuromuscular Disease: A Systematic Review

**DOI:** 10.3390/jcm12216834

**Published:** 2023-10-29

**Authors:** Lisa Pomp, Jeroen Antonius Lodewijk Jeneson, W. Ludo van der Pol, Bart Bartels

**Affiliations:** 1Child Development and Exercise Center, Wilhelmina Children’s Hospital, University Medical Center Utrecht, 3584 CX Utrecht, The Netherlands; 2Department of Neurology, Brain Center Rudolf Magnus, University Medical Center Utrecht, 3584 CX Utrecht, The Netherlands

**Keywords:** exercise training, neuromuscular diseases, biomarkers, imaging, MRI, MRS, EMG, ultrasound, NIRS, systematic review

## Abstract

Exercise therapy as part of the clinical management of patients with neuromuscular diseases (NMDs) is complicated by the limited insights into its efficacy. There is an urgent need for sensitive and non-invasive quantitative muscle biomarkers to monitor the effects of exercise training. Therefore, the objective of this systematic review was to critically appraise and summarize the current evidence for the sensitivity of quantitative, non-invasive biomarkers, based on imaging and electrophysiological techniques, for measuring the effects of physical exercise training. We identified a wide variety of biomarkers, including imaging techniques, i.e., magnetic resonance imaging (MRI) and ultrasound, surface electromyography (sEMG), magnetic resonance spectroscopy (MRS), and near-infrared spectroscopy (NIRS). Imaging biomarkers, such as muscle maximum area and muscle thickness, and EMG biomarkers, such as compound muscle action potential (CMAP) amplitude, detected significant changes in muscle morphology and neural adaptations following resistance training. MRS and NIRS biomarkers, such as initial phosphocreatine recovery rate (V), mitochondrial capacity (Q_max_), adenosine phosphate recovery half-time (ADP t_1/2_), and micromolar changes in deoxygenated hemoglobin and myoglobin concentrations (Δ[deoxy(Hb + Mb)]), detected significant adaptations in oxidative metabolism after endurance training. We also identified biomarkers whose clinical relevance has not yet been assessed due to lack of sufficient study.

## 1. Introduction

The health impact of endurance exercise training lies in the improvement of cardiovascular and skeletal muscle function by improving oxidative metabolism through increasing capillary density, muscle blood flow, mitochondrial size and density, and enzyme activity in skeletal muscle. In addition, endurance training leads to decreased heart rate at rest, increased cardiac stroke volume, and increased total blood volume. The combination of these adaptations increases the aerobic work capacity and anaerobic threshold [1,2,3,4,5]. Resistance or strength training leads to increased power, increased lactic threshold, increased maximal oxygen uptake, and decreased body fat, and promotes neural adaptations, skeletal muscle hypertrophy, and strength gains in healthy people [6,7,8,9,10]. To monitor exercise training efficacy, non-invasive methodologies and various quantitative indices have been used as ‘biomarkers’ of muscle health. For example, magnetic resonance imaging (MRI), magnetic resonance spectroscopy (MRS), surface electromyography (sEMG), ultrasound, and near-infrared spectroscopy (NIRS) have been used in addition to conventional clinical outcome measures, such as peak force and peak torque [11,12,13,14,15,16]. MRI is a non-invasive method for creating detailed images of organs and tissues using magnetic fields and radiofrequency. The difference between various types of tissues can be seen based on the body’s natural magnetic properties [17]. MRS non-invasively measures the concentrations of tissue metabolites, providing information on a wide range of biochemical processes in the body in vivo [18]. In muscles, phosphorus MRS (^31^P MRS) is mostly used to monitor muscle energy metabolism [19]. sEMG assesses the myoelectric output of a muscle, such as the intensity of muscle contraction, the myoelectric expression of muscle fatigue, and the recruitment of motor units [20,21]. Ultrasound utilizes sound waves to produce non-invasive internal images, and NIRS measures the tissue oxygen status [22].

Exercise may represent a therapy approach for patients with neuromuscular diseases (NMDs) characterized by reduced muscle strength and endurance, but its application is limited by the heterogeneity of NMD, safety concerns, and limited insights into which strategies work best. NMDs constitute a heterogeneous group of probably more than 500 genetic and acquired disorders characterized by dysfunction of the peripheral neuromuscular system, i.e., motor-neurons, nerves, neuromuscular junctions, and muscles [23]. Many NMDs are progressive, causing increasing levels of disability due to muscular weakness, exercise intolerance, and fatigue [24]. This may result in a vicious cycle of incremental inactivity, leading to further deconditioning, loss of muscular strength, and increased fatigability. Since there is no cure for the majority of known NMDs, the primary aim of treatment is to maintain function and mobility [11], to which exercise therapy could contribute. 

The efficacy of exercise training approaches in patients with NMD is largely unknown. Most NMDs are rare, which hampers the execution of larger studies. Moreover, exercise intervention protocols often differ regarding the frequency, intensity, type, and time (FITT factors), and functional outcome measures may not always be suitable for the training intervention [24,25]. 

These inherent limitations, when exploring the efficacy of exercise training in NMD, could at least partially be addressed using sensitive biomarkers [26]. Biomarkers would allow the quantitative assessment of training efficacy within the spectrum of NMDs, caused by changes in the physiology or function of the motor unit [27]. In addition, biomarkers can be used as a tool to maintain the safety of an intervention program by monitoring potential complications, such as muscle inflammation or edema [28]. 

The objective of this systematic review is, therefore, to critically appraise and summarize the evidence for the sensitivity of available, quantitative, non-invasive imaging and electrophysiological biomarkers used to measure the effect of a physical exercise training intervention in people with an NMD. 

## 2. Methods

The review was not registered, and the review protocol was not prepared. 

### 2.1. Eligibility Criteria

We included studies in which the study design fulfilled all the following criteria: (1)The patients studied had a confirmed diagnosis of neuromuscular disease (we excluded patients with diagnoses of diabetic neuropathies, compression, or entrapment neuropathies,, radiculopathy, thoracic outlet syndrome, or complex regional pain syndrome).(2)The study involved a longitudinal exercise intervention of more than 6 weeks, the minimal period for neural adaptations.(3)The key outcomes were measured by MRI, MRS, sEMG, ultrasound, or NIRS.(4)The study included a comparison with non-exercise intervention controls within NMD patients, and/or a comparison before and after the intervention within NMD patients, and/or a comparison with healthy controls.

We excluded animal studies, case reports, and studies with invasive measurement techniques. 

### 2.2. Search Strategy

We searched PubMed, EMBASE, CINAHL, and Cochrane databases to identify all original articles concerning human neuromuscular disease studies with imaging and/or electrophysiological biomarkers for exercise therapy. We included articles up until 9 January 2023. The search strategy included three main components: (1) ‘neuromuscular disease’; (2) ‘exercise therapy’ OR ‘exercise’; and (3) ‘magnetic resonance imaging (MRI)’ OR ‘magnetic resonance spectroscopy (MRS)’ OR ‘ultrasonography’ OR ‘electromyography (EMG)’ OR ‘Spectroscopy, Near-Infrared (NIRS)’. The terms consisted of title abstract keywords and indexed subject headings (MeSH and Emtree terms in the databases). The full search string can be found in Appendix A. We imported all retrieved studies to EndNote 20 software (Thomson Reuters, NY, USA) and removed duplicates.

### 2.3. Study Selection and Data Extraction

Two of the authors (L.P. and B.B.) screened article references independently against the inclusion criteria. First, title and abstract (TIAB) screening were performed using Rayyan (www.rayyan.ai, Qatar Computing Research Institute (QCRI), Doha, Qatar. URL accessed on 9 January 2023). Second, two researchers (L.P. and B.B.) performed a full-text screening independently. Any disagreements regarding the inclusion or exclusion of a particular publication were resolved by discussion. One author (L.P.) performed the data extraction of the following study characteristics from eligible records:Method: date of the study and study type.Participants: number, age, gender, disease, and baseline characteristics.Interventions: intervention (frequency, intensity, type, time), comparison, concomitant treatments, and excluded treatments.Outcomes: primary and secondary outcomes specified and collected, and time points. *p*-values were provided when given.

### 2.4. Risk of Bias

Two authors (L.P. and B.B.) independently appraised the study quality. We used the Cochrane Risk of Bias 2.0 (RoB 2) tool to assess the risk of bias in randomized controlled trials (RCTs) [29]. Furthermore, we assessed the non-randomized controlled trials with The Risk Of Bias In Non-randomized Studies-of Interventions (ROBINS-I) assessment tool [30]. We rated the pre–post studies using the Quality Assessment Tool for Before–After (Pre–Post) Studies with No Control Group (12 items) of the National Institutes of Health (NIH). We subsequently classified the RCTs and pre–post studies as low, with some concerns or a high risk of bias using the guidance provided within the appointed tools. The non-randomized controlled trials were classified as low, moderate, serious, and critical risk of bias using ROBINS-I. 

### 2.5. Best Evidence Synthesis

We identified imaging and electrophysiological biomarkers from the studies with a low risk of bias or with some concerns and excluded the studies with a high or serious risk of bias. To gain insight into the sensitivity of the biomarkers, we compared the effect of endurance and/or resistance training on functional outcomes per study with the effect measured by imaging and electrophysiological biomarkers. 

## 3. Results

### 3.1. Study Selection

The bibliographic search strategy retrieved a total of 1619 articles, including 812 from EMBASE, 751 from PubMed, 34 from CINAHL, and 22 from Cochrane (Figure 1). After the removal of duplicates, we screened 1390 unique articles based on titles and abstracts. During this title and abstract screening, we excluded 1366 studies, leaving 24 articles for full-text screening. After full-text screening, we excluded seven more articles because the article appeared to be abstract only (n = 6) or the intervention appeared to have been shorter than 6 weeks (n = 1). Finally, we included 17 studies in this systematic review. 

### 3.2. Study Characteristics and Quality Assessment

The 17 included studies featured a wide variety of study designs, neuromuscular diseases, intervention characteristics, and outcome biomarkers. The results of the quality assessment of the 17 included records are summarized in Table 1. We included five RCTs, one non-randomized controlled study, and eleven pre–post studies with no control. According to the ROB-2 quality assessment tool for RCTs, we rated three studies [26,32,33] as having a high risk of bias, one record [34] as having some concerns with the risk of bias, and one record [35] as a low risk of bias. We rated one study [36] as having a serious risk of bias according to the ROBIN-1 quality assessment tool. Furthermore, according to the NIH Pre–post quality assessment for non-controlled studies, we rated two studies [37,38] as poor study quality, and nine records [39,40,41,42,43,44,45,46,47] as fair study quality.

To elaborate on the factors causing a risk of bias, none of the studies reported whether the sample size was sufficiently large to provide statistical power in the findings and all studies lacked or did not report blinding of both their patients and assessors. Furthermore, the studies lacked an interrupted time-series design of the outcome measures. For the RCTs, the risk of bias was determined mainly due to deviations from the intended interventions. Another identified risk of bias was non-adherence and the lack of appropriate analysis to estimate the effect of the non-adherence in four studies. For more detail, we refer to the full report of the quality assessment in Appendix A.

Table 1 summarizes the main patient characteristics of the studies. Overall, the 17 studies included 212 patients representing 11 different NMDs. Population cohorts consisted of 50 patients with mitochondrial myopathy (MM), 38 patients with Charcot–Marie–Tooth disease (CMT), 26 patients with myasthenia gravis (MG), 24 patients with Polymyositis (PM), 18 patients with Duchenne muscular dystrophy (DMD), 15 patients with dermatomyositis (DM), 10 patients with chronic unspecified non-metabolic myopathies (NMM), 9 patients with facioscapulohumeral muscular dystrophy type 1 (FSHD1), 9 patients with myotonic dystrophy (MD), 7 patients with McArdle disease (McA), and 6 patients with Post Polio Syndrome (PPS). The age range was between 8 and 80 years.

### 3.3. Exercise Intervention Characteristics

Table 2 presents the exercise characteristics of the studies regarding the FITT factors. The range of the intervention duration was between 8 weeks and 6 months, with the number of sessions varying between 20 and 130 sessions. The exercise interventions were performed at moderate and high intensity with, e.g., maximal heart rate at 80%. The studies also varied regarding the type of exercise; most studies focused on endurance exercise (n = 8), some on both resistance (or strength) exercise and endurance exercise (n = 5), and a few on resistance exercise alone (n = 5).

### 3.4. Best Evidence Synthesis

#### 3.4.1. Biomarkers Measured by MRI

Eight records used MRI-derived biomarkers to investigate the effect of exercise training as can be seen in Table 3. 

From the studies with the best evidence (some concerns and low risk of bias), the MRI biomarkers measured were muscle volume or area [35,39,41,43], fat infiltration [35,37,38,41], inflammatory changes [37,38,41], and muscle damage [47]. 

Trenell et al. (2005) performed endurance training for 12 weeks in patients with MM and observed significant effects on endurance-based functional measures, as shown in Appendix A [43]. MRI showed significant gains of muscle volume and muscle maximum area after the training program. 

Burns et al. (2017) performed resistance training for 6 months in patients with CMT and observed a significant effect on strength dorsiflexion but not on gait nor Charcot–Marie–Tooth disease Pediatric Scale (CMTPedS) score [35]. However, no significant effects were seen on the MRI biomarkers of muscle volume or fat infiltration. Similarly, Spector et al. (1996) executed resistance training for 10 weeks in patients with PPS and observed significant effects of resistance-based functional measures [39]. No endurance-based functional measures were present in this study. Again, no significant effects were seen on the MRI biomarker muscle maximum area. Tollbäck et al. (1999) studied the effect of a resistance training program lasting 12 weeks in patients with MD and observed a significant effect on resistance-based functional measures as well [41]. No significant effects were seen on muscle maximum area or fat infiltration. In addition, the safety of training was supported by the lack of an increase in inflammatory changes. Lott et al. (2021) used only MRI to measure the muscle damage biomarker as a safety measure. After a resistance training program of 12 weeks in patients with DMD, they detected no significant detrimental effect on muscle morphology [47].

#### 3.4.2. Biomarkers Measured by MRS

Five studies evaluated metabolic biomarkers measured with MRS as presented in Table 3. The best evidence metabolic biomarkers were resting phosphocreatine (PCr) [43], resting adenosine diphosphate (ADP) [43], resting pH [43], PCr hydrolysis during exercise [43], pH fall during exercise [43], end exercise ADP [43], initial PCr recovery rate (V) [42,43], mitochondrial capacity (Q_max_) [42,43], pH recovery rate [43], capacity of the proton efflux system (d*E*/d(pH fall) rate) [43], muscle oxidative capacity (ADP t_1/2_) [36,40], and PCr/beta-nucleoside triphosphate (β-NTP) ratio [34]. 

After the 8-week endurance training program in MM patients (1998), significant effects were seen on endurance-based functional measures (Appendix A). Furthermore, this study showed that endurance exercise training improved muscle oxidative capacity (ADP t1/2) significantly. Similarly, Taivassalo et al. (2001) performed endurance training in patients with MM for 14 weeks and observed significant effects on endurance-based functional measures, such as work capacity. This study showed significant changes in the MRS biomarkers V and Q_max_ [42]. Trenell et al. (2005) performed endurance training for 12 weeks in patients with MM and again observed significant V and Q_max_ effects, but not other MRS biomarkers [43]. Chung et al. (2007) showed significant effects in endurance-based functional measures, but not in resistance-based functional measures after a combined endurance and resistance exercise intervention of 6 months in PM and DM patients [34]. 

#### 3.4.3. Biomarkers Measured by EMG

sEMG was used to measure electrophysiological biomarkers (fatigue indices, root mean square, coactivation percentage, mean frequency, and compound muscle action potential (CMAP)), as can be seen in Table 3. 

Mhandi et al. (2007) [44] performed interval endurance training for 24 weeks in patients with CMT and observed significant effects on both endurance-based and resistance-based functional measures, as can be seen in Appendix A. However, no significant effects were seen on the sEMG biomarkers fatigue indices, root mean square, coactivation percentage, or mean frequency. 

Westerberg et al. (2018) demonstrated significant effects on endurance-based and resistance-based functional measures, such as isometric muscle force in the quadriceps but not in the biceps brachii, after 12 weeks of combined endurance and resistance training in patients with MG [46]. The CMAP of the quadriceps, but not biceps, changed significantly after the training program. 

#### 3.4.4. Biomarkers Measured by Ultrasound

The thickness of the muscle [33,46], the pennation angle [33], and the fascicle length [33] were measured by ultrasound in the best-evidenced studies. 

Westerberg et al. (2018) [46] demonstrated a significant increase in the thickness of the rectus femoris and vastus intermedius, but not the biceps brachii, after a combined endurance and resistance training lasting 12 weeks in patients with MG. Endurance-based and resistance-based functional measures, such as the 30 s chair stand test, improved significantly as well.

#### 3.4.5. Biomarkers Measured by NIRS

NIRS was only used in one study to detect training effects. The biomarker measured with NIRS in the study by Porcelli et al. (2016) was the micromolar change in deoxygenated hemoglobin and myoglobin concentrations (Δ[deoxy(Hb + Mb)]) [45]. After an endurance training program of 12 weeks in patients with MM and McA, significant effects were seen on endurance-based and resistance-based functional measures for both diseases. Significant effects on Δ[deoxy(Hb + Mb)] were also seen in patients with MM and McA. 

### 3.5. Quality of Evidence

As we considered the studies in this systematic review to be heterogeneous with regard to the study population, methodological quality, FITT factors, and assessment of functional outcomes, we refrained from statistically pooling the data. 

## 4. Discussion

This review includes 17 studies regarding the effect of exercise training on imaging and electrophysiological biomarkers in 242 people with NMDs [26,32,33,34,35,36,37,38,39,40,41,42,43,44,45,46,47]. Exercise duration ranged from 8 to 26 weeks, using endurance, resistance, or both types of training. The studies included 28 different biomarkers that were measured by five techniques, MRI, MRS, sEMG, ultrasound, and NIRS, with MRI being applied most often (n = 8 records). The risk of bias was variable, but mostly with some concerns (n = 10) or high risk (n = 6). This was mainly caused by small sample sizes, lack of blinding of participants and outcome assessors, only two time-point measurements, and no appropriate analysis when deviations from the intended interventions occurred. Although other reviews only consider studies with a low risk of bias [48] due to the fundamental nature of exercise interventional studies in a rare disorder with small sample sizes and inadequate blinding of participants and/or assessors, we included studies with some concerns of bias as well in the best evidence synthesis. 

All studies on endurance training or combined training showed a significant increase in endurance-based functional measures. Only Trenell et al. (2005) used MRI to measure the effect of endurance training on muscle volume and muscle maximum area. The result was a significant increase in both biomarkers [43]. However, this was unexpected as endurance training does not induce hypertrophy [49]. It is possible that the intensity of the endurance program also implied the recruitment of type II muscle fibers. Another possibility would be that muscle hypertrophy was caused by mechanical tension, metabolic stress, or muscle damage [50]. MRS biomarkers were investigated mainly after endurance training, as MRS detects changes in muscle energy metabolism. V, Q_max,_ and ADP t_1/2_ improved significantly after endurance training, and therefore appear to be sensitive for detecting exercise effects. Mhandi et al. (2007) did not show significant effects in sEMG biomarkers after endurance training [44]. However, trends of improvements in these outcomes were observed in the fatigue indices and coactivation percentage. With NIRS, the effect of endurance training was also observed. Δ[deoxy(Hb + Mb)] was increased significantly in MM and McA, which implies an improvement in skeletal muscle oxidative metabolism. This is in line with our hypothesis, as endurance training causes metabolic and thus oxidative adaptations.

All studies on resistance training or combined training showed a significant increase in resistance-based functional measures, except Chung et al. (2007) [34]. The hypothesis is that resistance training leads to a decrease in body fat, neural adaptations, and skeletal muscle hypertrophy in addition to improved functional outcomes, such as strength gains. However, in the studies by Burns (2017), Spector (1996) and Tollbäck (1999), no significant changes in muscle morphology were seen with MRI after resistance training [35,39,41]. In healthy subjects, muscle hypertrophy can already be detected with MRI after 3 weeks of resistance training [51]. Therefore, it can be stated that muscle maximum area or volume could be sensitive biomarkers for detecting training effects; however, in the NMDs studied, these changes are negligible. Interestingly, Westerberg et al. (2018) did observe a significant increase in muscle thickness after resistance training using ultrasound [46]. This study did not use MRI, and we did not identify other comparative studies. It is, therefore, unclear whether ultrasound is more sensitive than MRI for detecting changes. The CMAP amplitude represents the sum of motor unit action potentials in the muscle, influenced by factors such as muscle fiber number and size, and the synchronization of the muscle fibers depolarization. The CMAP amplitude showed a significant increase in the quadriceps after a combined training program. This is in line with the study by Molin et al. (2016), where CMAP amplitude was higher in the resistance-trained healthy population in comparison to the not-trained healthy population. CMAP amplitude correlated with isometric muscle strength [52]. 

This systematic review revealed a great number of imaging and electrophysiological biomarkers with little uniformity to assess exercise training. It is important to establish a standardized set of outcomes based on the exercise intervention to enable comparisons of data across studies. This will ensure that key outcomes are consistently measured and reported, facilitating more accurate and meaningful interpretations of exercise research results. Moreover, it is key to investigating the biomarkers complementary to the exercise type for future research.

It is hard to speculate whether the biomarkers would indicate a similar result with a different disease population or a slightly different intervention program. For example, one study showed a significant effect on muscle volume [43] that other records did not [35,39]. Additionally, it was demonstrated that the results could be dependent on specific muscle types. To give an example, in the study by Westerberg et al. (2018), the exercise intervention only had a significant effect on the rectus femoris and vastus intermedius and not on the biceps brachioradialis as far as muscle thickness was concerned, even though the biceps brachioradialis was also trained [46]. Therefore, it is difficult to compare the results of the shared biomarkers between different patient populations, exercise interventions, genders, and ages. 

Most studies investigating the effect of training do not use any imaging or electrophysiological biomarkers, but rather, they focus mainly on functional outcomes [11,48]. Significant training effects can be measured by these functional outcomes (Appendix A). All 11 best-evidenced studies showed significant improvements in functional measures. Therefore, we can state that the functional tests are a valid method for detecting training effects. However, functional tests do not explain fundamentally how and where these improvements were made. The use of multiple functional tests would allow us to distinguish which muscles or parts of the body improved. Moreover, the great advantage of more fundamental imaging and electrophysiological biomarkers is that they can clarify which part of the motor unit function improved or which part of the motor unit is misfunctioning in a specific disease or person. Moreover, the training intervention can be modified because of these outcomes to offer patients a more specified training intervention. For wider clinical implications, to be able to enhance personalized and supervised exercise training for patients, these significant evidence-based results need to be translated into clinical care. 

A striking finding in the study by Janssen et al. (2016) is that the fat fraction increase normalized per year was significantly decelerated after the endurance training intervention compared to usual care [26]. In the functional tests, no significant effect of the training intervention was observed. Although this paper contains a high risk of bias, it presents the potential of the biomarkers, as muscle adaptations after an exercise intervention could be measured earlier than with a functional test. In the other studies of this systematic review, a significant training effect was measured with functional outcomes when a significant effect was measured with imaging or electrophysiological biomarkers. 

A limitation of most studies included is that they only compare the results before and after the training intervention or with healthy controls. More measurement time points could provide more knowledge on the development of imaging or electrophysiological biomarkers. In addition, with progressive neuromuscular diseases, a stabilization of the biomarkers could also mean the exercise intervention has a significant effect. The best ways to investigate whether the patients stabilized or improved are to either compare the intervention group with the usual care group or to include additional pre-intervention measurement time points. Therefore, biomarkers which do not show a significant difference could still be clinically relevant for measuring the exercise training effect. Unfortunately, we cannot draw conclusions about which biomarkers would still be clinically relevant due to a lack of information on the natural history of the diseases. 

For future studies of endurance training, it is recommended that V, Q_max_, ADP t_1/2,_ and/or Δ[deoxy(Hb + Mb)] are used as biomarkers to detect changes in oxidative adaptations. For resistance training, muscle maximum area, thickness, and/or CMAP amplitude are recommended for monitoring training effects. These biomarkers showed a significant change after a training protocol in various NMDs. For monitoring muscle damage or inflammation, the safety biomarkers measured with MRI are recommended, as in certain NMDs, training programs still involve risks. 

In the future, imaging and electrophysiological biomarkers can play a significant role in guiding decision-making, informing treatment strategies, and improving patient outcomes in several ways. For instance, imaging and electrophysiological techniques can aid in diagnosing NMDs and assessing their severity. Biomarkers provide objective data on, e.g., the extent of motor neuron loss, muscle atrophy, and neuromuscular dysfunction. Furthermore, the mentioned biomarkers can help clinicians choose the most appropriate treatment for patients with NMDs, for example, in spinal muscle atrophy (SMA). SMA is characterized by the deterioration of the spinal cord α-motor neurons, resulting in severe muscle weakness and wasting [53]. The treatment of SMA has undergone significant changes with the introduction of the first effective disease-modifying treatments. The next phase in the ongoing care of SMA patients involves the development of combined therapies that include SMN replacement treatment and additional approaches aimed at preserving and enhancing the entire motor unit, encompassing motor neurons, neuromuscular junctions, and muscles, throughout various stages of the disease [54,55,56]. Clinicians can use biomarkers to determine the level of muscle wasting, helping to decide whether the patient is a candidate for (innovative) SMN-enhancing drugs and/or supportive care interventions. Not only could these validated sensible biomarkers be used to monitor training effects, but also to measure the effect of medication treatments or combinatorial treatments. Biomarkers can serve as objective endpoints to measure treatment efficacy, allowing for more efficient and informative trials. Biomarkers can clarify where, exactly, the adaptations occur in the motor unit when treatment mechanisms are unknown. Moreover, these biomarkers could detect treatment response in a patient before clinical symptoms become apparent and could help decision-making to continue or to end a treatment intervention, to reduce costs and patient burden. Lastly, biomarkers could help to design personalized treatment interventions. By assessing the unique biomarker profile of each patient, clinicians can tailor treatment strategies to address specific needs and vulnerabilities. This individualized approach can lead to more effective and improved outcomes. 

## 5. Conclusions

This systematic review critically evaluated the use of non-invasive imaging and electrophysiological biomarkers to assess the effect of physical exercise training in patients with NMDs. We identified a variety of biomarkers which were measured with techniques such as MRI, MRS, sEMG, ultrasound, and NIRS. The biomarkers V, Q_max_, ADP t_1/2,_ and Δ[deoxy(Hb + Mb)] detect significant adaptations in oxidative metabolism after endurance training when significant effects were observed on endurance-based functional measures. Furthermore, muscle maximum area, thickness, and CMAP amplitude were able to detect significant muscle morphology and neural adaptations after resistance training, when significant effects were observed on resistance-based functional measures as well. Although functional measures are more sensitive than these biomarkers for detecting training effects, the added value of these biomarkers is that they explain the more fundamental adaptations in the muscle that cause the functional effects. Therefore, these biomarkers are recommended for monitoring exercise effects alongside functional measures. The other identified biomarkers cannot be rejected, as they may still be clinically relevant. With the biomarkers, a more specified training program can be designed for the various NMDs. 

## Figures and Tables

**Figure 1 jcm-12-06834-f001:**
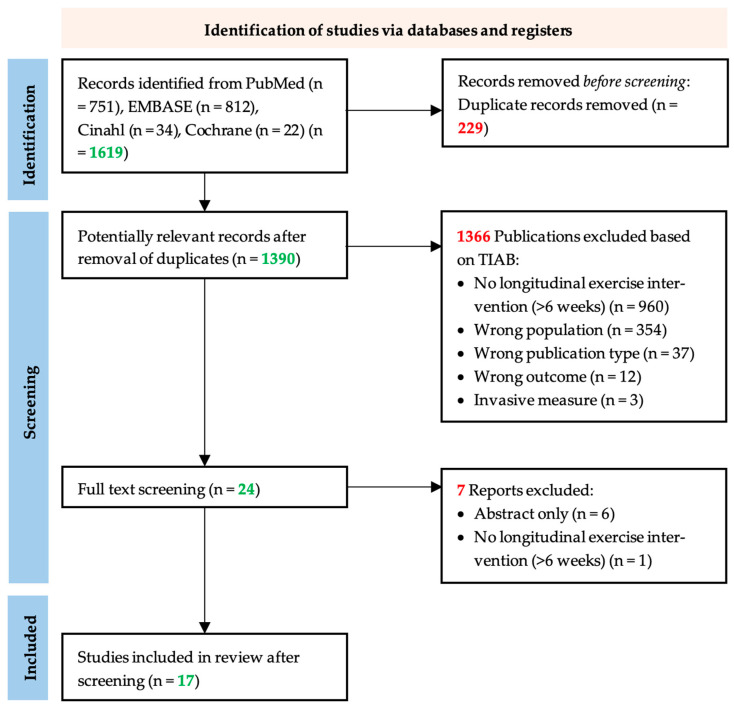
PRISM flow chart of the study selection process. The systematic search in EMBASE, PubMed, CINAHL, and Cochrane yielded 1390 unique publications. After title and abstract screening, articles were screened by full text, after which, seven articles were excluded based on the exclusion criteria. Following the addition of 0 articles from references of included studies, information from 17 articles was extracted for the systematic review [31].

**Table 1 jcm-12-06834-t001:** Overview of literature, results of the quality assessment, and patient characteristics.

First Author	Year	Study Type	Risk of Bias	NMD	Number of Participants	Gender (M/F)	Age (y) ^4^
Chung [34] ^2^	2007	RCT	Some concerns	PM and DM	n = 18; 12 PM, 6 DM	3/15	Mean 50
Janssen [26] ^3^	2016	RCT	High	FSHD1	n = 9	7/4	56 ± 15
Burns [35] ^3^	2017	RCT	Low	CMT	n = 30	16/14	11.5 ± 3.3
Rahbek [32]	2017	RCT	High	MG	n = 15	7/8	55.6 ± 17.2
Bulut [33] ^3^	2022	RCT	High	DMD	n = 10	10 M	7.9 (7.2–8.7)
Taivassalo [36] ^3^	1999	N-RCT	Serious	MM and NMM	n = 24; 14 MM, 10 NMM	9/15	14–63
Spector [39]	1996	Pre–post	Some concerns	PPS	n = 6	5/1	53 ± 7
Taivassalo [40]	1998	Pre–post	Some concerns	MM	n = 10	4/6	36 ± 9
Alexanderson [38]	1999	Pre–post	High	PM and DM	n = 10; 5 PM, 5 DM	2/8	27–60
Tollbäck [41]	1999	Pre–post	Some concerns	MD ^1^	n = 9	2/7	37 ± 8.6
Alexanderson [37]	2000	Pre–post	High	PM and DM	n = 11; 7 PM, 4 DM	3/8	23–80
Taivassalo [42]	2001	Pre–post	Some concerns	MM	n = 10	4/6	39.3 ± 9.5
Trenell [43]	2005	Pre–post	Some concerns	MM	n = 10	3/7	42 ± 14
Mhandi [44]	2007	Pre–post	Some concerns	CMT	n = 8	8 M	23–45
Porcelli [45]	2016	Pre–post	Some concerns	MM and McA	n = 13; 6 MM, 7 McA	4/2, 3/4	51 ± 16, 41 ± 13
Westerberg [46]	2018	Pre–post	Some concerns	MG	n = 11	5/6	60 ± 18
Lott [47]	2021	Pre–post	Some concerns	DMD	n = 8	8 M	9.3 ± 0.8

^1^ The abbreviation for myotonic dystrophy is usually DM; however here, DM is used for dermatomyositis, therefore MD is used instead. ^2^ Only the placebo group without creatine supplementation. The placebo group would be given lactose. ^3^ Only the group with a training intervention is shown. ^4^ The age range and ±SD below is as given in the study. Abbreviations: NMD = neuromuscular disease; M = male; F = female; y = years; RCT = Randomized controlled pre–post study; N-RCT = Non-randomized controlled pre–post study; Pre–post = Pre–post study with no control; PM = Polymyositis; DM = Dermatomyositis; FSHD1 = Facioscapulohumeral Muscular Dystrophy Type 1; CMT = Charcot–Marie–Tooth disease; MG = Myasthenia Gravis; DMD = Duchenne Muscular Dystrophy; MM = Mitochondrial Myopathy; NMM = Chronic Non-metabolic Myopathies; PPS = Post polio Syndrome; MD = Myotonic Dystrophy; McA = McArdle Disease.

**Table 2 jcm-12-06834-t002:** Exercise intervention FITT characteristics.

Study	Frequency	Intensity	Type	Time (Min)
Chung [34]	5 d/w program and 7 d/w walk for 6 m	Moderate	Strength and endurance	15 + 15 walk
Janssen [26]	3 d/w for 16 w	50 to 65% of HR reserve	Endurance (cycling)	30 + 5–10 walk, increased by 1 min daily
Burns [35]	3 d/w for 6 m	Resistance at 50 to 70% of the repetition max; Sham at <10% of the repetition max	Strength or sham	25
Rahbek [32]	5 d/2 w for 8 w (20 sessions)	Moderate to high	(1) Endurance (cycling); (2) Progressive resistance (full-body program)	(1) 3 × 10–12 with 3 min rest; (2) Various
Bulut [33]	3 d/w for 12 w	60% of max HR	Endurance (cycling)	40
Taivassalo [36]	3–4 d for 8 w	70–85% of HRR	Endurance (treadmill)	20–30
Spector [39]	3 d/w for 10 w	75% of the three-repetition maximum	Strength	NR
Taivassalo [40]	3–4 d for 8 w	60–80% of HRR	Endurance (treadmill)	20–30
Alexanderson [38]	5 d/w for 12 w	Moderate	Strength and endurance	15 + 15 walk
Tollbäck [41]	3 d/w for 12 w	80% of one-repetition maximum	Strength	~10 min (program)
Alexanderson ^1^ [37]	<3 d/w program and 5 d/w walk for 12 w	Moderate	Strength and endurance	15 + 15 walk
Taivassalo [42]	3–4 d for 14 w (50 sessions)	70–80% of max HR	Endurance (cycling)	30–40
Trenell [43]	3 d/w for 12 w	70–80% of their age-predicted max HR	Endurance (cycling)	30
Mhandi [44]	3 d/w for 24 w	HR at 80% max aerobic power	Interval-endurance (cycling)	45 (program)
Porcelli [45]	4 d/w for 12 w	65–70% of max HR	Stretching and endurance (cycling)	15 stretching + 30–45 endurance
Westerberg [46]	2 d/w for 12 w	Minimum to 80% max HR	Endurance, strength and balance	90 (program)
Lott [47]	3 d/w for 12 w	Mild-moderate (50% MVC and after 6 w 60% MVC)	Strength	90

^1^ In the study by Alexanderson et al. (2000), corticosteroids were used together with the exercise intervention [37]. Abbreviations: d = days; w = weeks; min = minutes; m = months; HR = heart rate; HRR = heart rate reserve; NR = not reported; MVC = maximal voluntary contraction.

**Table 3 jcm-12-06834-t003:** Biomarkers and their baseline and end intervention results.

	First Author	Measurement Day	Baseline	End Intervention
MRI biomarkers				
Muscle volume	Burns [35]	0, 182	1.2 ± 0.3 (scaled score)/control: 1.2 ± 0.4	1.1 ± 0.3 (scaled score)/control: 1.1 ± 0.3 (*p* = 0.24)
	Trenell [43]	0, 84	11.8 ± 1.5 dm^3^	**12.8 ± 1.6 dm^3^ (*p* < 0.05)**
Muscle maximum area	Trenell [43]	0, 84	464 ± 65 cm^2^	**497 ± 70 cm^2^ (*p* < 0.05)**
	Tollbäck [41]	0, 84	4090 ± 591 mm^2^	4154 ± 585 mm^2^
	Spector [39]	0, 70	n.a.	Not significantly changed
Fat infiltration	Burns [35]	0, 182	0.1 ± 0.1 (scaled score)/control: 0.1 ± 0.1	0.1 ± 0.1 (scaled score)/control: 0.1 ± 0.1 (*p* = 0.25)
	Alexanderson [38]	0, 84	n.a.	Increased amount of fat in n = 1
	Alexanderson [37]	0, 84	No fat infiltration	No change
	Tollbäck [41]	0, 84	Fatty replacement in n = 4	n.a.
Fat fraction	Janssen [26]	0, 112	32 ± 36%	**Increase in fat fraction normalized per year: 2.9% (*p* = 0.03) (Significantly decelerated compared with UC (6.7%))**
Inflammation	Alexanderson [38]	0, 84	n.a.	No increased muscle inflammation
	Alexanderson [37]	0, 84	Inflammation in n = 3	Inflammation in n = 2, no signs of inflammation in n = 5
	Tollbäck [41]	0, 84	n.a.	No increased muscle inflammation
Muscle damage	Lott [47]	0, 84	KE = 47 ± 5 ms and KF = 44.5 ± 3 ms	KE = +2.3% (SD 3.6) and KF = +0.4% (SD 4.6)
^31^P MRS biomarkers				
Resting PCr	Trenell [43]	0, 84	29 ± 1 mmol.L^−1^	27 ± 1 mmol.L^−1^
Resting ADP	Trenell [43]	0, 84	28 ± 5 μmol.L^−1^	34 ± 5 μmol.L^−1^
Resting pH	Trenell [43]	0, 84	7.04 ± 0.01	7.06 ± 0.01
PCr hydrolysis during exercise	Trenell [43]	0, 84	13 ± 1 mmol.L^−1^	12 ±1 mmol.L^−1^
pH fall during exercise	Trenell [43]	0, 84	−0.3 ± −0.1	−0.3 ± −0.1
End exercise ADP	Trenell [43]	0, 84	57 ± 5 μmol.L^−1^	59 ± 10 μmol.L^−1^
V	Trenell [43]	0, 84	0.4 ± 0.1 mmol.L^−1^.min^−1^	**0.7 ± 0.1 mmol.L^−1^.min^−1^ (*p* < 0.05)**
	Taivassalo [42]	0, 98	10.0 ± 4.7 mmol.L^−1^.min^−1^	**14.1 ± 5.5 mM/min (*p* < 0.05)**
Q_max_	Trenell [43]	0, 84	20 ± 3 mmol.L^−1^.min^−1^	**26 ± 2 mmol.L^−1^.min^−1^ (*p* < 0.05)**
	Taivassalo [42]	0, 98	12.6 ± 6.0 mmol.L^−1^.min^−1^	**17.2 ± 6.5 mM/min (*p* < 0.05)**
Initial pH recovery rate	Trenell [43]	0, 84	5 ± 2 mmol.L^−1^.min^−1^	5 ± 2 mmol.L^−1^.min^−1^
d*E*/d(pH fall) rate	Trenell [43]	0, 84	20 ± 3 mmol.L^−1^.min^−1^(pH unit)^−1^	19 ± 1 mmol.L^−1^.min^−1^(pH unit)^−1^
ADP t_1/2_	Taivassalo [40]	0, 56	* 3.50 ± 0.56 min/** 0.68 ± 0.48 min	*** 0.59 ± 0.18 min/** 0.40 ± 0.22 min (*p* < 0.04)**
	Taivassalo [36]	0, 56	MM: 1.27 ± 1.30 min/NMM: 0.35 ± 0.23 min	**MM: 0.48 ± 0.33 min (*p* < 0.01)**/NMM: 0.28 ± 0.15 min
PCr/β-NTP ratio	Chung [34]	0, 91, 182	4.03 ± 0.29	4.05 ± 0.31
sEMG biomarkers				
Fatigue indices	Mhandi [44]	0, 84, 168	−8.9 ± 7.2%	−6.9 ± 5.5%
Root mean square	Mhandi [44]	0, 84, 168	58%	58%
Coactivation	Mhandi [44]	0, 84, 168	11%	6.5%
Mean frequency	Mhandi [44]	0, 84, 168	68 Hz	63 Hz
Integrated EMG	Rahbek [32]	0, 56	PRT: 10.9 ± 4.2 μV/ET: 8.5 ± 2.4 μV	PRT: 13.0 ± 7.0 μV (*p* = 0.31)/ET: 7.0 ± 3.4 μV (*p* = 0.52)
CMAP amplitude	Westerberg [46]	0, 84	Quadriceps: 4.5 ± 2.6 mV/BB: 5.5 ± 2.1 mV	**Quadriceps: 5.3 ± 2.8 mV (*p* = 0.016)**/BB: 4.6 ± 1.3 mV (*p* = 0.63)
Ultrasound biomarkers				
Thickness	Westerberg [46]	0, 84	Rectus femoris: 19.6 ± 5.6 mm/vastus intermedius: 18.0 ± 5.8 mm/BB: 33.3 ± 6.5 mm	**Rectus femoris: 23.0 ± 3.9 mm (*p* = 0.0098)/vastus intermedius: 22.0 ± 6.2 mm (*p* = 0.034)**/BB: 32.1 ± 6.2 mm (*p* = 0.11)
	Bulut [33]	0, 84	Vastus lateralis: D—2.1 ± 0.5 cm/ND—2.2 ± 0.4 cm (control: D—2.1 ± 0.2 cm/ND—2.2 ± 0.2 cm)	Vastus lateralis: D—2.3 ± 0.6 cm/ND—2.3 ± 0.4 cm (control: D—2.4 ± 0.5 cm (*p* *** = 0.6)/ND—2.5 ± 0.5 cm (*p* *** = 0.5))
Pennation angle	Bulut [33]	0, 84	Vastus lateralis: D—18.9 ± 3.7°/ND—19.6 ± 3.4° (control: D—17.7 ± 3.3°/ND—18.4 ± 2.8°)	Vastus lateralis: D—19.7 ± 5.3°/ND—18.5 ± 6.3° (control: D—18.8 ± 2.3° (*p* *** = 0.9)/ND—20.4 ± 2.6° (*p* *** = 0.3))
Fascicle Length	Bulut [33]	0, 84	Vastus lateralis: D—7.0 ± 1.5 cm/ND—6.7 ± 1.5 cm (control: D—7.5 ± 1.6 cm/ND—6.8 ± 1.7 cm)	Vastus lateralis: D—7.0 ± 1.3 cm/ND—7.5 ± 1.7 cm (control: D—7.6 ± 0.9 cm (*p* *** = 0.8)/ND—7.3 ± 0.6 cm (*p* *** = 0.6))
NIRS biomarkers				
Δ[deoxy(Hb + Mb)]	Porcelli [45]	0, 84	MM: 22.0 ± 6.7% of ischemia/McA: 23.4 ± 6.2% of ischemia	**MM: 34.2 ± 5.9% of ischemia (*p* < 0.05)/McA: 40.6 ± 7.2% of ischemia (*p* < 0.05)**

* results of two patients with a putative nuclear DNA mutation. ** results of the remaining eight patients. *** between group comparison *p*-value (*p* < 0.05 is significant). **Bold indicates a significant difference between baseline and end intervention.** Abbreviations: d = days; MRI = magnetic resonance imaging; ^31^P MRS = phosphor magnetic resonance spectroscopy; sEMG = surface electromyography; NIRS = near-infrared spectroscopy; PCr = phosphocreatine; ADP = adenosine phosphate; pH = potential of hydrogen; V = initial PCr recovery rate; Q_max_ = mitochondrial capacity; E = rate of net proton efflux; ADP t_1/2_ = adenosine phosphate recovery half-time; β-NTP = beta-nucleoside triphosphate; CMAP = compound muscle action potential; Δ[deoxy(Hb + Mb)] = micromolar changes in deoxygenated hemoglobin and myoglobin concentrations; KE = knee extensors; KF = knee flexors; MM = Mitochondrial Myopathy; NMM = Chronic Non-metabolic Myopathies; PRT = progressive resistance training; ET = endurance training; BB = biceps brachii; D = dominant leg; ND = non-dominant leg; McA = McArdle disease. n.a. = Not applicable.

## Data Availability

No new data were created or analyzed in this study. Data sharing is not applicable to this article.

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
