# Peer review of "Electrophysiological and Imaging Biomarkers to Evaluate Exercise Training in Patients with Neuromuscular Disease: A Systematic Review"

_jcm, 2023, doi:10.3390/jcm12216834_

Round 1

Reviewer 1 Report

This systematic review is a relevant report and it is methodology well done

This reviewer has a few comments-

In the abstract, it is mentioned electromyography and surface electromyography, since the former can be invasive and only the latter is approached, it seem that this duplication is incorrect.

In introduction, all NMD are rate, not true, diabetic PNP is quite common, for example.

In methods, spinal cord injury is not a NMD, should not be mentioned an excluded NMD.

In results, “nearly all studies lacked blinding…” better to detail de numbers. Tables 1 and 2 can be merged in a single one. Tables 1, 2 and 3 should keep the same authors order from top to bottom to facilitate reading and understanding. The description of the different techniques (MRI, US, etc) are not fitted to results, please move to introduction. The section 3.6 is not results but a comment, it should be moved to Discussion.

In discussion, the authors should acknowledge that CMAP amplitude derives from muscle fiber number and diameter, but also from the synchronization of the muscle fibers depolarization, which is influenced by sarcolemma conduction velocity.

The titles wording in the reference list are not separated.

Reviewer 2 Report

The authors presented a nice manuscript regarding the use of imaging and neurophysiological biomarkers which could be correlated (in different ways) with exercise training status in patients with different neuromuscular disorders. Some points to be evaluated by the authors at this point include:

1. We know that some authors describe Post Polio Syndrome as Post Polio Muscular Atrophy. However, as the terminology with syndrome refers to a classic description in the medical literature my suggestion is to consider only the description as a syndrome. 

2. There was a marked number of individuals with myopathies if compared to other neurogenic disorders. Have the authors any idea of why there were no studies related for example with ALS or SMA? 

3. I think the authors did very well including studies which had minor concerns regarding potential bias, as stated in line 334. It is complex to try to perform specific studies in the rare disease population trying to blind participants or assessors. In fact, most attempts to have a potential double-blinded context are not properly addressed by authors and this certainly gives rise to biased data at some point.

4. Is the near-infrared spectroscopy a validated biomarker in the study of neuromuscular diseases and training programs or is it mainly a hypothetical biomarker? I ask this because the near-infrared spectroscopy is not a widely available resource in most centers.
